Effects of reindeer grazing and recovery after cessation of grazing on the ground-dwelling spider assemblage in Finnish Lapland

Saikkonen Teemu 1
Vahtera Varpu 1
Koponen Seppo 1
Suominen Otso otsosuo@utu.fi 2
1 Zoological Museum, Biodiversity Unit, University of Turku , Turku , Finland
2 Kevo Subarctic Research Institute, Biodiversity Unit, University of Turku , Turku , Finland
Björk Robert
Electronic publication date: 2019 Jul 17
Publication date: 2019
Volume: 7
Electronic Location ID: e7330
Received 2018 Mar 9; Accepted 2019 Jun 20
Copyright: ©2019 Saikkonen et al.
Copyright year: 2019
Copyright holder: Saikkonen et al.
License: This is an open access article distributed under the terms of the Creative Commons Attribution License, which permits unrestricted use, distribution, reproduction and adaptation in any medium and for any purpose provided that it is properly attributed. For attribution, the original author(s), title, publication source (PeerJ) and either DOI or URL of the article must be cited.
License URL: https://creativecommons.org/licenses/by/4.0/

Keywords: Rangifer tarandus, Grazing, Spider assemblage, Thomisidae, Lycosidae, Lichen, Linyphiidae

Funding: Academy of Finland 105964 This work was supported by the Academy of Finland (postdoctoral project 105964) funding for Otso Suominen. The funders had no role in study design, data collection and analysis, decision to publish, or preparation of the manuscript.

==============================
The effect of reindeer Rangifer tarandus L. grazing on the ground-dwelling spider assemblage in Northern Finland was studied. Changes in species richness, abundance and evenness of spider assemblages were analyzed in relation to changes in vegetation and environmental factors in long term grazed and ungrazed sites as well as sites that had recently switched from grazed to ungrazed and vice versa. Grazing was found to have a significant impact on height and biomass of lichens and other ground vegetation. However, it seemed not to have an impact on the total abundance of spiders. This is likely caused by opposing family and species level responses of spiders to the grazing regime. Lycosid numbers were highest in grazed and linyphiid numbers in ungrazed areas. Lycosidae species richness was highest in ungrazed areas whereas Linyphiidae richness showed no response to grazing. Four Linyphiidae, one Thomisidae and one Lycosidae species showed strong preference for specific treatments. Sites that had recovered from grazing for nine years and the sites that were grazed for the last nine years but were previously ungrazed resembled the long term grazed sites. The results emphasize the importance of reindeer as a modifier of boreal forest ecosystems but the impact of reindeer grazing on spiders seems to be family and species specific. The sites with reversed grazing treatment demonstrate that recovery from strong grazing pressure at these high latitudes is a slow process whereas reindeer can rapidly change the conditions in previously ungrazed sites similar to long term heavily grazed conditions.

Introduction

Wild reindeer have been a natural part of the northern boreal ecosystems in Eurasia since the last ice age, but in northern Fennoscandia and many parts of northern Russia they have been replaced by reindeer husbandry with a sustained high density of semi-domesticated reindeer. Contrary to the reindeer herding practices in Norway and Sweden, in Finland, semi-domesticated reindeer herds can no longer have the traditional long migration between their separate summer and winter ranges. The Finnish reindeer herding area is divided into 57 herding districts defined by legislation (Suominen & Olofsson, 2000). These districts are relatively small (about 1,000 to 5,000 km2) compared to the herding areas in Norway and Sweden and prevent long distance migration. At present, modern herding practices force reindeer herds in Finland to graze in restricted areas. It is especially detrimental to the lichen dominated winter pastures, if reindeer are forced to use them at summertime since that leads to strong trampling impact. Kumpula, Colpaert & Nieminen (2000) showed that the condition of lichen rich winter pastures is not explained by reindeer density in relation to the total land area of a herding district but by the density of reindeer in relation to the area of lichen ranges. Nowadays with the inclusion of supplementary feeding which prevents natural mortality owing to shortage of food, the grazing pressure in relation to the carrying capacity of pastures can rise excessively high (Kojola, Helle & Aikio, 1991; Helle & Kojola, 1993; Evans, 1996; Kumpula et al., 1997; Suominen & Olofsson, 2000). Local herders’ cooperatives are trying to save their best winter grazing areas from overgrazing by a pasture rotation system (Kumpula, Fielitz & Colpaert, 1999), but the success of this is limited.

In the boreal ecosystem, the semi-domesticated reindeer (Rangifer tarandus tarandus L.) can be seen as an ecosystem engineer or modifier as well as a disturbance. The ecosystem engineer concept (Jones, Lawton & Shachak, 1994; Jones, Lawton & Shachak, 1997) implies that, in addition to impacts of reindeer through trophic interactions, they control the physical properties of the ecosystem as well. The concept of disturbance, according to the intermediate disturbance hypothesis (IDH) (Connell, 1978), implies that the species diversity is at its peak when the level of disturbance is intermediate. IDH has often been applied to the impact of herbivores on vegetation diversity. Both roles stem from grazing behavior and are not mutually exclusive.

Sustained high density of deer has been shown to have a negative impact on vegetation and this impact is further cascading to the whole ecosystem including animal assemblages (Côté et al., 2004). Several studies have demonstrated that the changes caused by large herbivores can have an impact on species composition and abundance of several other animal taxa through various direct and indirect mechanisms (reviewed in Suominen & Danell, 2006; Foster, Barton & Lindenmayer, 2014). At the current constantly high densities reindeer herds have substantial influence on the environment; irrespective of whether reindeer considered to be a native part of the ecosystem or a disturbance due to high densities and is lack of natural pasture rotation migration. For instance, grazing can change, sustain and create habitats (Putman, 1994), accelerate (Kielland, Bryant & Ruess, 1997; Augustine & McNaughton, 1998; Stark et al., 2000; Stark, Strömmer & Tuomi, 2002; Olofsson, Stark & Oksanen, 2004) and even decelerate nutrient flux by affecting the detritus food web via changes in microclimate (Väre, Ohtonen & Mikkola, 1996; Pastor & Cohen, 1997; Augustine & McNaughton, 1998; Olofsson et al., 2001). In addition, grazing has an impact on the community structure and species diversity of vegetation (e.g.,  Väre, Ohtonen & Mikkola, 1996; Pastor & Naiman, 1992; Pastor & Cohen, 1997; Milchunas, Lauenroth & Burke, 1998; Suominen, 1999) and can influence the post-disturbance succession rate of vegetation (Oksanen, Moen & Helle, 1995; Kielland, Bryant & Ruess, 1997; Helle et al., 1998). The impacts of reindeer on vegetation and soil properties as well as on the interaction between soils, vegetation and climate change have been studied a lot in recent years (e.g.,  Eskelinen, Kaarlejärvi & Olofsson, 2017; Kaarlejärvi, Eskelinen & Olofsson, 2017; Ylänne et al., 2017; Egelkraut et al., 2018; Maliniemi et al., 2018). Especially in winter ranges of reindeer in boreal forests, which are typically dry pine forests with Cladonia-lichen dominated bottom layer vegetation, it has been shown that the main impact of reindeer is a strong sift from thick lichen carpets to much less bottom and field layer vegetation (e.g.,  Väre, Ohtonen & Mikkola, 1996; Suominen & Olofsson, 2000; Köster et al., 2013). In addition to biomass and composition of forest floor vegetation this then has cascading impacts on other ecosystem properties such as tree recruitment, fungal communities, litter decomposition (Köster et al., 2013; Köster et al., 2015; Santalahti et al., 2018). Changes observed in vegetation can subsequently mediate the impact of grazing on invertebrate partners in a community (Owen-Smith, 1987; Decalesta, 1994; Baines, Sage & Baines, 1994; Bromham et al., 1999; Suominen, 1999; Wardle et al., 2001; Suominen et al., 2003; Suominen & Danell, 2006; Foster, Barton & Lindenmayer, 2014).

The forest floor spider community is greatly influenced by the habitat’s abiotic conditions, such as moisture, light and temperature (Uetz, 1991) as well as by the structure of the vegetation and other three dimensional habitat microhabitat features (Colebourn, 1974; Hartley & Macmahon, 1980; Koponen, 1995), and the presence of reindeer can change these conditions and these changes can have an impact on the availability of prey for spiders. Since spiders act both as abundant predators in the invertebrate community and as important prey for numerous forest animals, the changes in spider assemblage can further cascade through the food web via numerous direct and indirect ways.

In this study, we examine the impact of reindeer grazing and trampling on the ground-dwelling spider fauna in a special area with four different types of reindeer grazing history. It is also unique in the way that moose (Alces alces) is present in both sites with and without reindeer grazing. Our hypothesis is that reindeer through the changes caused in vegetation height, structure and biomass as well as in the abiotic conditions has an impact on the assembly structure of spiders. More specifically, we hypothesize that sites with recently reversed reindeer grazing status would have intermediate environmental conditions and assembly structure compared long term grazed and ungrazed sites. Following the IDH assumptions we expected a peak in diversity of spider assemblages at intermediate grazing treatments compared to long term grazed or ungrazed pastures.

Methods

Study area

The study area (Fig. 1) was in eastern Finnish Lapland, in Raja-Jooseppi Inari (68°28′N, 28°28′E) where semi-domesticated reindeer have been traditionally herded by tradition by Sámi people for centuries on the Finnish territory. The study site was in dry lichen dominated pine heath of typical reindeer winter pasture on Lappi reindeer herding district’s eastern boarder. The district has got an area of 4,396 km2, and its maximum number of overwintering reindeer is 8,000 individuals. On the Russian side of the national border there are no wild or semi-domesticated reindeer in the area close to the border. The study sites were in the restricted border zone on the Finnish side of the border. We had a permission for the work from the Finnish Boarder Guard (permit number 2717/2015).

Figure 1 Map of Finland showing the study area.

A reindeer fence along the Finnish-Russian border was erected in the 1940s, and in 1997 a new fence crossing the old one was built in our study area. With time, the old fence was torn down so that herds could enter untouched pastures. The junction of the new fence and the position of the old fence forms a cross, where four different grazing regions are distinctly present: (i) UGraz—continuously ungrazed from 1940. (ii) Graz—continuously grazed from at least 1940. (iii) (new ungrazed) NUGraz—ungrazed from 1997, but previously grazed. (iiii) (new grazed) NGraz—grazed from 1997 but previously ungrazed.

Reindeer fences prevent reindeer from entering the Russian side, where reindeer are locally extinct due to historical causes. Moose, however, occur on both sides and are capable of crossing the fences. This gives a unique opportunity to solely study the grazing effects of reindeer.

Pitfall trapping

Spiders were collected using pitfall traps. The traps were plastic cups 170 ml in volume each with a mouth diameter of 70 mm. The killing/preserving fluid was a mix of polypropylene glycol (60%) and tap water, with a hint of detergent added. With pitfall traps it is possible to passively collect a large proportion of actively moving fauna, and being abundant and actively moving creatures, spiders are frequently encountered in pitfalls. A comparison of spider sampling techniques has proved pitfall trapping to maintain its catching efficiency most constantly in time (Churchill & Arthur, 1999).

Trapping design

A total of 36 trap lines, each consisting of five traps positioned three meters apart, were placed in six plots in the vicinity of the reindeer fence junction in an area of 150 × 500 m. Each plot included three treatments: the upper region of the fence crossing had Graz, NGraz and UGraz; the lower region had Graz, NUGraz and UGraz. Each treatment within a plot had two trap lines, one five meters from the fence and another 20 m from the fence. All the trap lines were parallel to the fence. Duo to the possible fence effect, i.e., packing of the deer against the fence and thus higher grazing and trampling impact (e.g., Oksanen, 1978; Olofsson et al., 2001), we tested prior to the further analyses whether the lines with different distance to the fence differed from each other (paired t-test or Wilcoxon signed-ranks test).

The trapping period was two months from early June to early August 2005 and due to the short summer of northern boreal regions it practically covered the entire growing season. The emptying interval was one month.

Identification of mature specimens was done to species level. As juveniles often lack distinctive species characters (Norris, 1999) they were identified to family level.

Vegetation and environmental factors

Species composition and height of field and bottom layer vegetation was evaluated with a 50 × 50 cm square situated in the middle of each trap line. Dry biomass of the vegetation and soil moisture of soil samples were collected in mid-July, and measured in the laboratory. In each trap line, temperature was measured at the depth of 5 cm in the soil (June 7th 2005, to nearest 0.1 °C). Since the impacts of reindeer on Cladina lichen dominated forest floor vegetation have been documented in several other studies (e.g., reviews by Suominen & Olofsson, 2000; Bernes et al., 2015), and the main focus of the present study is on the spider community, we present the vegetation results here only as major functional groups pooled over species: lichens, mosses and vascular plants. In our study sites, all other vascular plants than ericaceous dwarf shrubs, were too rare to be statistically tested, and all these shrubs had similar response to treatments. Thus, the result on vascular plants is in fact equal to the response of these shrubs.

Data analysis

Data was analyzed using SAS Enterprise Guide 3.0, EcoSim (Gotelli & Entsminger, 2001), and Canoco for Windows 4.5 (Ter Braak & Smilauer, 1998). Equality of variances was tested with Levene’s test. Normality of residuals was tested with the Shapiro–Wilk test. Variables that were not normally distributed were analyzed with the nonparametric Kruskal-Wallis χ2-test or Spearman rank correlation (see the table captions). Since many key variables such as spider abundances and plant biomasses could not be analysed with parametric tests the effect of treatments on even those response variables that fulfilled the requirements of parametric tests were also tested with simple one-way ANOVA. In the statistical analyses we used trapping lines as sampling units (observations) i.e., data over the individual traps on each line was combined. The vegetation and physical variables were also measured at trapping line level.

Interactions between environmental variables and species were analyzed using linear regression. The Detrended Correspondence Analysis (DCA) ordination method was used to analyze community structure. The two options “downweighting of rare species” and “trend correction by segments” of Canoco for Windows were turned on.

As the number of species and the number of individuals per sample are often strongly correlated, the number of individuals caught in each trap line was adjusted by rarefaction (see Gotelli & Colwell, 2001) to make trap lines of dissimilar catch size comparable.

Results

The distance of the trap line from the fence, i.e., “fence effect”, was found to have no effect on spiders and vegetation (paired t-test or Wilcoxon signed-ranks test, p > 0.1) and in the subsequent analysis this classifying factor was removed.

General abundance and species numbers

A total of 4,225 spider specimens were collected during the whole sampling period. Of these, 3,426 specimens were collected from early June till early July and 804 from early July to early August. Mature spiders that were identified to species level numbered 3,241 individuals, and 984 specimens were juveniles and therefore identified to family level only (see Supplement for a complete list of specimens). In total 73 spider species were caught, representing 11 families (Table 1).

Table 1 List of spider species found in the study.

Araneidae	
Hypsosinga albovittata (Westring, 1851)	
Cercidia prominens (Westring, 1851)	
Dictynidae	
Hackmania prominula (Tullgren, 1948)	
Gnaphosidae	
Gnaphosa lapponum (L. Koch, 1866)	
Gnaphosa montana (L. Koch, 1866)	
Gnaphosa muscorum (L. Koch, 1866)	
Gnaphosa sticta (Kulczynski, 1908)	
Haplodrassus signifer (C. L. Koch, 1839)	
Haplodrassus soerenseni (Strand, 1900)	
Micaria alpina (L. Koch, 1872)	
Zelotes subterraneus (C. L. Koch, 1833)	
Hahnidae	
Hahnia ononidum (Simon, 1875)	
Linyphiidae	
Agyneta cauta (O. P.-Cambridge, 1902)	
Agyneta conigera (O. P.-Cambridge, 1863)	
Agyneta gulosa (L. Koch, 1869)	
Agyneta subtilis (O. P.-Cambridge, 1863)	
Agyneta trifurcata (Hippa & Oksala, 1985)	
Bolepthyphantes index (Thorell, 1856)	
Ceratinella brevipes (Westring, 1851)	
Centromerus arcanus (O. P.-Cambridge, 1873)	
Cnephalocotes obscurus (Blackwall, 1834)	
Decipiphantes decipiens (L. Koch, 1879)	
Diplocentria bidentata (Emerton, 1882)	
Diplocentria rectangulata (Emerton, 1915)	
Hilaira herniosa (Thorell, 1875)	
Macrargus rufus (Wider, 1834)	
Maso sundevalli (Westring, 1851)	
Micrargus herbigradus (Blackwall, 1854)	
Minyriolus pusillus (Wider, 1834)	
Moebelia penicillata (Westring, 1851)	
Mughiphantes cornutus (Schenkel, 1927)	
Neriene clathrata (Sundevall, 1830)	
Oreoneta sinuosa (Tullgren, 1955)	
Palliduphantes antroniensis (Schenkel, 1933)	
Pocadicnemis pumila (Blackwall, 1841)	
Porrhomma pallidum (Jackson, 1913)	
Scandichrestus tenuis (Holm, 1943)	
Semljicola latus (Holm, 1939)	
Sisicus apertus (Holm, 1939)	
Tapinocyba pallens (O. P.-Cambridge, 1872)	
Tenuiphantes alacris (Blackwall, 1853)	
Tenuiphantes mengei (Kulczynski, 1887)	
Tenuiphantes tenebricola (Wider, 1834)	
Tibioploides arcuatus (Tullgren, 1955)	
Tibioplus diversus (L. Koch, 1879)	
Walckenaeria antica (Wider, 1834)	
Walckenaeria capito (Westring, 1861)	
Walckenaeria cuspidata (Blackwall, 1833)	
Walckenaeria dysderoides (Wider, 1834)	
Walckenaeria karpinskii (O. P.-Cambridge, 1873)	
Walckenaeria obtusa (Blackwall, 1836)	
Walckenaeria unicornis (O. P.-Cambridge, 1861)	
Zornella cultrigera (L. Koch, 1879)	
Liocranidae	
Agroeca proxima (O. P.-Cambridge, 1871)	
Lycosidae	
Acantholycosa lignaria (Clerck, 1757)	
Alopecosa aculeata (Clerck, 1757)	
Alopecosa pinetorum (Thorell, 1856)	
Pardosa eiseni (Thorell, 1875)	
Pardosa hyperborea (Thorell, 1872)	
Pardosa lasciva (L. Koch, 1879)	
Pardosa lugubris (Walckenaer, 1802)	
Pardosa palustris (Linnaeus, 1758)	
Pardosa sphagnicola (Dahl, 1908)	
Philodromidae	
Thanatus formicinus (Clerck, 1757)	
Salticidae	
Evarcha falcata (Clerck, 1757)	
Neon reticulatus (Blackwall, 1853)	
Therididae	
Robertus lividus (Blackwall, 1836)	
Theonoe minutissima (O. P.-Cambridge, 1879)	
Thomisidae	
Ozyptila arctica (Kulczynski, 1908)	
Xysticus audax (Schrank, 1803)	
Xysticus luctuosus (Blackwall, 1836)	
Xysticus obscurus (Collett, 1877)	

In terms of specimen numbers (number of spider individuals, abundance), the most common families were wolf spiders (Lycosidae) and sheet weavers (Linyphiidae). Nearly one third of the mature spiders were wolf spider Pardosa eiseni or Alopecosa aculeata individuals. The twelve most common species comprised 76% of the total number of spiders (Table 2). Eight species were encountered only once.

Table 2 Twelve most common species and their specimen numbers.

Species	Pardosa eiseni	Alopecosa aculeata	Pardosa palustris	Ozyptila arctica	Hilaira herniosa	Pocadicnemis pumila	
n	622	344	276	235	200	159	
Species	Agyneta gulosa	Pardosa hyperborea	Walckenaeria karpinskii	Palliduphantes antroniensis	Hahnia ononidum	Micrargus herbigradus	
n	148	118	117	103	80	67	

Vegetation and environmental variables by treatment

Total lichen dry biomass (Fig. 2) and lichen height differed significantly between treatments (df = 5, χ2= 26,65, p<0.0001; df = 5, χ2=43,18, p<0.0001) Soil moisture did not differ between treatments, but soil temperature did (df = 5, F = 4.09, p = 0.0212) (Fig. 2). The impact of treatment on response variables is shown in Table 3.

Figure 2 The differences between treatments in soil temperature (°C at the depth of 5 cm), soil moisture (top layer of mineral soil), and total lichen biomass (dry mass g/0.25 m2) (mean ± SD).

The treatments are Graz—long term grazed area, Ugraz—long term ungrazed, NGraz—grazed last 9 years, but previously ungrazed, NUgraz—ungrazed last 9 years, but previously grazed (the number (1) in treatments Graz and Ugraz refers to sites that act as a reference to the NGraz treatment and number (2) to those that act as reference to the NUgraz treatment).

Table 3 Impact of treatment on response variables (mean and SD, or estimate with 95% confidence limits).

Variables that fulfill the assumptions for a parametric test were tested with ANOVA and the F-test (DF = 5). Spider abundances and lichen biomass were tested with the non-parametric Kruskal-Wallis χ2-test (DF = 5). Species richness for pooled families, Lycosidae and Linyphiidae, are rarefied values. Due to the low number of species and specimens, species richnesses of Thomisiidae and Gnaphosidae are absolute values without rarefaction. Site-level richness (α-diversity) is the rarefied number of species per trapping line. γ-diversity is the rarefied number of species per treatment (with 95% confidence limit). Bold font indicates variables with statistically significant (p < 0.05) treatment impact.

	Mean (SD)						F(χ2)	P	
	UGraz (1)	UGraz (2)	Graz (1)	Graz (2)	NUGraz	NGraz			
SPIDER RICHNES									
α	26.8 (4.1)	28.5 (2.3)	23.13 (2.1)	27.12 (5.9)	28.8 (6.0)	21.4 (2.8)	1.53	0.251	
γ	41.5 (3.1)	49.3 (1.5)	38.7 (4.1)	47.0 (0.2)	46.0 (0.5)	36.9 (2.5)	–	–	
Lycosidae	4.1 (0.2)	6.1 (0.4)	3.6 (0.3)	4.5 (0.7)	4.7 (1.5)	3.6 (0.8)	4.37	0.017	
Linyphiidae	11.5 (1.4)	9.8 (1.4)	11.2 (1.7)	10.2(3.1)	11.2 (1.7)	12.6 (2.0)	0.72	0.623	
Thomisidae	1.7 (0.6)	2.3 (0.6)	2.3 (0.6)	3.3 (0.6)	2.3 (0.6)	1.3 (0.6)	4.27	0.018	
Gnaphosidae	2.6 (0.6)	1.3 (0.6)	2.6 (0.6)	3.0 (1.7)	3.3 (1.5)	3.0 (1.0)	1.2	0.366	
SPIDER ABUNDANCE									
Total	203.7 (66.0)	166.7 (37.3)	225.3 (41.4)	135.6 (14.2)	144.6 (13.7)	206.0 (121)	5.63	0.344	
Lycosidae	93.3 (55.9)	22.3 (1.2)	123.3 (35.5)	47.7 (9.2)	40.0 (16.5)	147.7 (117.4)	12.09	0.034	
Linyphiidae	81.7 (3.8)	120.2 (35.3)	58.0 (3.5)	56.3 (9.3)	67 (7.2)	35.7 (10.0)	15.27	0.009	
Thomisidae	8.7 (7.6)	10.0 (6.1)	26.0 (4.0)	22.3 (5.5)	22.7 (3.1)	7.3 (2.1)	12.74	0.026	
Gnaphosidae	9.0 (2.6)	2.0 (1.0)	6.3 (3.5)	4.7 (3.1)	6.3 (2.5)	7.7 (6.7)	7.45	0.189	
P.antroniensis	10.7 (5.3)	22 (7.5)	0 (0)	0.7 (0.6)	0.7 (0.6)	0.3 (0.6)	24.41	<0.0001	
H. herniosa	10 (2)	47.3 (27.1)	1.7 (2.1)	0.7 (1.2)	5.3 (2.1)	1.7 (0.6)	7.88	0.002	
O. arctica	8 (5.3)	7.7 (5.5)	22.7 (2.9)	15.3 (8.1)	17.7 (2.9)	7 (1.7)	5.24	0.009	
A. gulosa	5.3 (6.8)	0.7 (1.2)	11.7 (6.7)	17 (13.1)	7.3 (6.8)	7.3 (8.5)	1.46	0.273	
ENV.VAR.									
Soil. °C	3.48 (0.45)	2.77 (2.07)	4.9 (0.56)	5.7 (0.80)	5.4 (1.84)	6.5 (0.24)	4.09	0.0212	
Moist %	0.18 (0.04)	0.16 (0.05)	0.15 (0.08)	0.09 (0.04)	0.13 (0.04)	0.10 (0.06)	1.27	0.338	
VEGET.HEIGHT									
Lichen	13.33 (1.32)	13.61 (0.78)	5.00 (1.12)	4.94(1.24)	7.22 (0.79)	5.10 (0.95)	43.18	<0.0001	
Moss	3.89 (2.15)	4.89 (1.76)	1.63 (0.74)	1.88 (0.78)	3.00 (1.87)	1.13 (0.23)	8.49	<0.0001	
Vascular plants	23.11 (8.08)	18.33 (3.84)	14.00 (2.35)	10.67 (3.31)	14.44 (4.25)	10.67 (2.35)	10.38	<0.0001	
VEGET.BIOM.									
Lichen	165.29 (68.14)	181.62 (49.12)	29.09 (9.84)	31.96 (11.96)	41.17 (10.50)	50.50 (17.27)	26.65	<0.0001	
Moss	8.36 (14.28)	11.23 (9.52)	4.29 (8.46)	8.62 (13.75)	21.40 (39.24)	2.38 (3.15)	0.77	0.580	
Vascular plants	43.40 (16.39)	35.45 (18.82)	28.09 (12.11)	36.64 (20.37)	34.45 (19.20)	23.43 (6.94)	1.09	0.384	

Dry biomasses of mosses or vascular plants were not significantly affected by grazing (Table 3). Vegetation height differed significantly between treatments for all functional groups i.e., lichens, mosses, and dwarf shrubs (practically all vascular plants in the field layer were dwarf shrubs) all being highest in ungrazed plots (Table 3).

Most common vascular plants were dwarf shrubs ling (Calluna vulgaris), lingonberry (Vaccinium vitis-idaea), crowberry (Empetrum nigrum) and bilberry (V. myrtillus). In addition to those, some graminoids (Poaceae), willow (Salix spp.), bog bilberry (Vaccinium uliginosum) and marsh Labrador tea (Rhododendron tomentosum, ex. Ledum palustre) grew on some sites. Among these plants there were three, which typically characterize more moist or even marshy sites, that were found only or almost exclusively in the long term ungrazed sites. Bog bilberry and Sphagnum-mosses were found only in the ungrazed sites where their occurrence frequencies were 36% and 20% of the studied plots. For marsh Labrador tea the occurrence frequency was 64% in ungrazed and 18% in grazed sites.

Soil moisture and soil temperature (df = 16, F = 8.18, p = 0.011), and lichen biomass and soil temperature (df = 16, F = 8.93, p = 0.009) correlated negatively, whereas no correlation was found between soil moisture and lichen biomass (Figs. 3A–3C).

Figure 3 The relationship between (A) lichen biomass (dry mass g/0.25 m2), (B) soil moisture (% top layer of mineral soil), and (C) soil temperature (°C at the depth of 5 cm).

Statistically significant linear regressions are included into the figures.

Impacts of grazing on spiders

Abundance

The total number of all spider individuals was similar between treatments, but at the family level there were differences (Table 3), correlations between spider families are shown in Table 4. Wolf spiders (Lycosidae) were most numerous in long grazed and recently grazed treatments; sheet weavers (Linyphiidae) were most abundant in ungrazed areas, while crab spiders (Thomisidae) showed a more complex yet family specific pattern (Table 3, Fig. 4).

Table 4 Spearman rank correlations of the abundances (n) and species richnesses (S) of spider families.

	Lycosidae n	Lycosidae S	Linyphiidae n	Linyphiidae S	
Lycosidae n	1				
Lycosidae S	−0.842**	1			
Linyphiidae n	−0.418	0.495*	1		
Linyphiidae S	0.712**	−0.628**	−0.203	1	
Others n	−0.294	0.376	−0.362	0.710**	
Others S	0.539*	-0.602**	0.221	−0.005	
Notes.

(*p < 0.05, **p < 0.01)(n = 18).

Others n and Others S mean the abundance and richness respectively of all other spider families than the one tested against.

Figure 4 Pooled family level abundances of spider individuals of the three most abundant spider families in different treatments (mean number of individuals per trapping line ± SD).

For the abbreviations of treatments, see the legend of Fig. 3.

Sheet weavers Palliduphantes antroniensis and Hilaira herniosa clearly occurred most often in ungrazed areas, whereas the sheet weaver Agyneta gulosa and the crab spider Ozyptila arctica were as clearly restricted to grazed areas (Fig. 5).

Figure 5 Abundances of spider individuals of four common species between different treatments (mean number of individuals per trapping line ± SD).

For the abbreviations of treatments, see the legend of Fig. 3.

Species richness

Rarefied trap line species richness (n = 123 for each trap line) did not differ significantly between treatments (Table 3). The rarefied wolf spider species richness (n = 21) was affected by treatments. The highest number of species (6, SD 0.4) was detected in ungrazed sites, the lowest in sites with a long history of grazing (3.5, SD 0.2).

Species richness of Thomisidae was also affected (F = 4.27, df = 5, p = 0.018) by treatments so that the highest richness was found in grazed areas. In sheet weavers the species richness showed no response to grazing.

Impact of vegetation and environmental factors on spiders

With increasing Cladina dry mass and height, the abundance of linyphiids and species richness of lycosids increased but the abundance of thomisids decreased (Table 5). Biomass of vascular plants did not correlate with the abundance or richness of any of the tested spider taxa (Spearman correlation n = 36, r < 0.27, P > 0.11). Moss biomass correlated negatively with Lycosidae richness (Spearman correlation n = 36, r =  − 0.460, P = 0.005), but not with other spider variables.

Table 5 The responses of spider family species richnesses (s) and abundances (n), as well as the abundances of the most numerous species to environmental variables.

Linear regression results are also included (*p < 0.05, **p < 0.01) (df = 16).

a)	Lichen bm	Soil moist. %	Soil temp. °C	b)	Lichen bm	Soil moist. %	Soil temp. °C	
	F	F	F		F	F	F	
Tot. species (S)	0,47	0,28	5,68*	A.aculeata	1,06	0,65	1,38	
Lycosidae S	3,64	0,33	10,71**	O.arctica	4,92*	0,1	2,02	
Linyphiidae S	0,41	2,31	0,02	H.ononidum	0,26	0,69	0,05	
Thomisidae S	1,09	0,11	0,01	P.antroniensis	37,17**	1,96	11,98**	
Gnaphosidae S	2,8	0,44	1,32	A . gulosa	6,28*	4,94*	6,52**	
Tot. specimens (n)	0,43	3,32	0	P. pumila	2,11	4,42*	1,87	
Lycosidae n	0,33	0,85	1,16	H.herniosa	17,7**	0,25	3,28	
Linyphiidae n	20,69**	1,09	8,09*	W.karpinskii	0,86	0,02	0,35	
Thomisidae n	6,54**	0,25	2,42	M.herbigradus	0,27	12**	6,57**	
Gnaphosidae n	0,11	3,954	0					
Notes.

S number of species

n number of specimen

Of the most common species Ozyptila arctica (Thomisidae) and Agyneta gulosa (Linyphiidae) decreased in abundance with increasing lichen dry mass and height, whereas Palliduphantes antroniensis (Linyphiidae) was clearly more abundant in areas with thick lichen cover. Also, Hilaira herniosa (Linyphiidae) showed a similar pattern to P. antroniensis. Other common species showed no response to changes in lichen dry mass or height (Table 5). With increasing soil moisture, the abundance of Gnaphosidae and the total species richness of spiders increased significantly. Micrargus herbigradus and Pocadicnemis pumila (Linyphiidae) were more abundant in moister areas, whereas A. gulosa numbers decreased with increasing moisture. The other common species did not respond significantly to soil moisture.

Higher soil temperature was connected to a decrease in linyphiid abundance, rarefied lycosid species richness and the total rarefied species richness of the spider assemblage in each treatment (Table 5). M. herbigradus and P. antroniensis decreased in abundance, while A. gulosa abundance was significantly higher in sites with warmer soil.

Correlations between spider families

Correlations between abundances and species richnesses of spider the two most numerous spider families, Lycosidae and Linyphiidae, are shown in Table 4. For the overall spider abundance and richness we used the abundance and richness of all other spider families than the one that was tested for, i.e., for Lycosidae all spiders except lycosids and for Linyphiidae all other spiders than linyphids. The abundance of wolf spider individuals correlates negatively with their own rarefied species richness and positively with the abundance of other spiders and the richness of linyphid spiders. The species richness of Lycosidae spiders correlated positively with the abundance of Linyphiidae and negatively with the richness other spiders and especially that of linyphids. High species richness of linyphids was associated with high abundance of spiders of other families. Earlier, Marusik & Koponen (2002) have found that rich lycosid fauna correlates with high overall species richness in the spider community as well analogous patterns. In our case the opposite was found when the lycosids themselves were excluded from the overall spied richness.

Multidimensional scaling

A DCA-ordination visualizing the effect of environmental factors and treatments on spider communities is presented in Fig. 6. Treatments are distinctly located on different sides of the graph depending on grazing history. Recently grazed and recently ungrazed treatments are grouped with long grazed areas on the first ordination axis, leaving the ungrazed area at the other end of the gradient. On the second ordination axis, no clear differences are detectable, except a more narrow distribution of ungrazed areas. Soil temperature and soil moisture varies between trap lines fairly independently of lichen dry biomass, but covary with each other to some extent.

Figure 6 DCA-ordination visualizing the effect of environmental factors and treatments on spider communities.

The symbols for treatments: long term grazed = □, sites where grazing status changed in 1997 = ◊, long term ungrazed = ○. The axes 1 and 2 explained 33.4 % and 16.7 % of the variation (relative eigenvalue portions 0.40 and 0.16 respectively).

The position of spider species on ordination axis 1 and 2 is presented in Fig. 7. Axis 1 mainly represents lichen biomass as in Fig. 6. Agyneta conigera and A. gulosa (Linyphiidae) are clearly detached from other species, indicating a preference for grazed areas over ungrazed. The lichen rich end of the first axis is dominated by H. herniosa (Linyphiidae), Robertus lividus (Theridiidae) and especially P. antroniensis (Linyphiidae).

Figure 7 The location of spider species on DCA-ordination axes 1 and 2.

Only species with more than 20% weight on the ordination are shown. The axes 1 and 2 explained 14.5 % and 9.3 % of the variation in the species data (relative eigenvalue portions 0.198 and 0.127, respectively).

On the second axis Pardosa hyperborea (Lycosidae) represents a distinct end point. H. herniosa and Minyriolus pusillus (Linyphiidae) amongst others are found at the opposite end. Xysticus obscurus (Thomisidae) is a species that did not react to the first axis, but reacted strongly to the second, while other species were situated somewhere between.

Discussion

Impact of grazing on spiders

Both family and species level responses to grazing were extensive in lycosids and linyphids, even though the combined abundance or species richness of all spiders did not differ between treatments.

As expected, wolf spiders were more abundant in grazed areas, and the response of sheet weavers was entirely the opposite. Several explanations for this pattern can be found. Wolf spiders chase their prey at ground level, thus making open habitat the optimal foraging ground. Wolf spiders are also relatively large in body size, making it hard to exploit thick, porous lichen carpet as a hunting ground. Also, because this group hunts actively by running, warmer and drier ground may be more suitable for them. Lower numbers of lycosids in ungrazed areas can also be explained by a lower number of suitable prey (Suominen, 1999), in addition to physically more demanding substrate. Linyphiidae spiders catch their prey by using webs, thus their habitat should comprise suitable anchoring sites for web structures. The abundance of web-building spiders has been found to be limited by both the availability of suitable web sites and prey availability (Minoshima et al., 2013 and references therein). Grazing has been shown to decrease both abundance and species richness of web-hunting spiders in Japan and in Kenya, by removing suitable web-building sites (e.g., Miyashita, Takada & Shimazaki, 2004; Warui et al., 2005; Takada et al., 2008; Minoshima et al., 2013). Some studies have suggested that the impact of large herbivores could also be mediated through the abundance of prey (Suominen et al., 2008; Takada et al., 2008) or direct physical disturbance to the spider webs (Foster et al., 2015). In several of these studies the method obtaining spider counts in the field has been based on direct observations of spider webs. This partially circularly reasoning methodology might overemphasize the importance of web building structures for the abundance of web-building spiders. Sweep net sampling in an area close to the present study site during the same summer did not show significant differences in the amount of insect prey (see Supplemental Information). Since the study area is typical winter pasture for reindeer it is unlikely that there would be substantial direct disturbance of the webs by reindeer during summer. A lack of suitable construction sites for webs and possibly also the different microclimate conditions (i.e., humidity and temperature) seem feasible explanations in this study.

Thomisidae abundance in both intermediate treatments strongly reflected the grazing history of the intermediate treatments. The observed similarity between recently ungrazed and continuously grazed sites could be explained by the slow recovery of vegetation. However, as continuously ungrazed and recently grazed sites also showed a resemblance, this hypothesis of prolonged succession seems incorrect.

Species richness of spiders

The response of rarefied wolf spider species richness to treatments was as expected. In the presumably favourable sites species richness was low and abundance high, whereas in low quality sites species richness was high and abundance low. This is well in line with the competitive exclusion theory, according to which strong competitors in good quality habitats can outcompete subdominants with a similar ecological niche, hence leading to lower species richness (Gause, 1934; Hardin, 1960). In suboptimal, ungrazed habitats, competitive exclusion is not effective as other factors than inter-species competition are more limiting. A high number of species could be seen as a sign of a more heterogeneous habitat, where more species can live together. However, a monotonous thick and dense carpet of lichen is not likely to be more heterogeneous than grazed areas or provide more suitable environments for ground dwelling wolf spiders.

Species richness within Linyphiidae family did not differ between treatments. As linyphiids are highly efficient in their dispersal capacities (ballooning method), they can respond efficiently to environmental changes. Yet, due to wind-driven and rather erratic dispersal, linyphiids are also often dispersed to suboptimal habitats, hence the species present in one spot might reflect chance only. Linyphiid abundance, however, can tell more about the actual quality of the habitat.

Spider species responses

A. gulosa and A. conigera, linyphid species found in grazed areas in high numbers, typically prefer areas in an early successional phase just after disturbance and are even considered pioneer species (Koponen, 2004; Koponen & Koneva, 2006). Occurrence of pioneer species in an area grazed continuously from at least the 1940s shows that grazing pressure is high enough to keep an area constantly in its early stage of succession. Linyphiidae spiders. P. antroniensis and H. herniosa are typical northern boreal bottom layer species (Koponen, 1999), and here strictly occurred in great abundance in ungrazed, reindeer-free areas.

Of the species that are not web-spinning, Xysticus obscurus (Thomisidae) was restricted to humid areas, but was rather infrequent in abundance and was previously found in multiple habitats (Koponen, 1977). P. hyperborea also indicated predominantly lichen rich, humid areas, but was also highly abundant in one site with low lichen biomass. The crab spider O. arctica was exceptional in its abundance by being most common in long term grazed sites as well as in newly ungrazed site.

Linear regression revealed the affinity of Micrargus herbigradus (Linyphiidae) to cool and moist sites. However, its occurrence was not affected by lichen biomass.

DCA-ordination did not group species into families but instead showed greatly differing environmental affinities, even within a genus. Hence, use of higher taxa than species to describe changes in environmental attributes can be misleading if the responses of the species contradict each other and the researcher is not utterly aware of the ecology of the focal species.

Importance of microclimate for spiders

In addition to causing differences in lichen biomass and height as well as other direct and indirect impacts on vegetation, grazing induces differences in microclimate, which may also explain the observed patterns in spider fauna (Koponen, Haukioja & Iso-Iivari, 1975; Samu, Sunderland & Szinetar, 1999; Frick, Nentwig & Kropf, 2007). Large, actively moving wolf spiders require a warmer habitat than small, relatively motionless web-hunting spiders. Due to their small body size, linyphiids are also more vulnerable to drying out. Inside and on top of the spongy, absorbent lichen carpet, the risk of desiccation is probably less important. Also, Riechert & Tracy (1975) showed that by building webs in physically optimal sites instead of sites maximizing prey catch, web-building spiders could significantly increase their fitness.

Impact of grazing on vegetation and environmental variables

As expected and shown by numerous previous studies (e.g., Väre, Ohtonen & Mikkola, 1996; Suominen & Olofsson, 2000; Köster et al., 2013), the main impact of reindeer on vegetation was a reduction in reindeer lichens. Reindeer significantly reduced both the biomass and the height of their main winter food, the Cladina lichens, that otherwise dominate ground layer vegetation in these forests. Grazing also raised the soil temperature in summer by removing the isolating and reflecting layer of white thick lichen. This differs from the results of the study by Köster et al. (2015). They did not find significant difference in soil temeprature in fairly similar conditions not so far away from our study site. We did not detect significant differences in soil moisture between treatments, and this is similar to the results of Köster et al. (2015). This may have been a result of small-scale topographical differences at the study site. However, occurrence of typical peatland species on the sandy soil of our study area like the spider Pardosa sphagnicola and plants like peat moss (Sphagnum spp.) and bog bilberry in the long term ungrazed area implies that there are actual differences in moisture levels between treatments. Also Pardosa hyperborea, being relatively abundant in our samples in ungrazed areas, is known to prefer moist conditions (Koponen, 1977), even to the extent of being one of the dominant species in peatlands of the boreal zone (Koponen, 2002).

The DCA ordination of the spider community split the treatments along two gradients: a first ordination axis that mainly represented lichen biomass and a second axis representing soil moisture and temperature. The latter, however, could not divide the species as clearly as the first. Most of the long grazed sites, along with intermediate treatments, were located at a relatively dry section of the gradient. The first axis clearly separated the ungrazed sites from the others, while variation within axis two was to some extent greater. Since the treatments with a recently altered grazing regime were grouped with long grazed treatment, eight years of protection from grazing appears not to be a long enough time to alter the composition of spider assemblage at the study site. Furthermore, areas with only a decade of grazing already resembled the long grazed areas in multiple ways.

Next to non-existent differences in vegetation between continuously grazed areas and areas with a decade of recovery from grazing were in contrast to our hypothesis emphasizing the importance of the level of grazing. Generally, harsh climatic conditions in northern boreal forests, leading to a slow succession rate and therefore minute detectable changes in vegetation in the time span of only one decade, may explain some of the findings. The study and modelling by Kumpula, Colpaert & Nieminen (2000) showed that if lichen has been grazed and trampled to extremely low level its recovery is much lower than that of more moderately depleted lichen stand. This might well be the case in our study area as well. Also, changing interspecific competition in the plant community due to changes in microclimate may slow down recovery from grazing (Olofsson, 2006). Patterns in Thomisidae abundance, however, imply that there might be real differences also between long grazed and intermediate treatments.

Conclusion

Based on our results, we could not find evidence of the hypothesized increase in spider diversity under intermediate grazing impact by reindeer in lichen dominated dry pine forests in the northern boreal zone, but the hypothesis cannot be rejected based on our data. Our data, of course, comes from only one year and a limited area, and as such is has some limitations. However, it seems rather that instead of being intermediate in relation reindeer impact the treatments where grazing impact had been reversed eight years ago were fairly similar to long term grazed habitats.

Focal species, sexes or even life stages can experience the same disturbance in dissimilar ways. However, in sites where one species was highly abundant, the species richness within that family was always lower than the species richness in sites of more even species distribution. This suggests active competitive exclusion.

Even though Linyphiidae and Lycosidae spiders responded to reindeer grazing in the same way as to clear-cut forest management in Pajunen et al. (1995), oversimplification of the detrimental effects of heavy grazing on the forest floor spider community should be avoided. The findings only support the existence of substantial habitat quality differences between grazed and ungrazed areas for spiders. Furthermore, it is vital to remember that foraging habits and thus the impact on vegetation also differ between large herbivore grazers (Huntly, 1991; Adler, Raff & Lauenroth, 2001). The distinct differences in feeding habits of reindeer between seasons can lead to completely different impacts of the same herbivore species, through winter grazing on ground lichens and summer grazing on grasses and deciduous leaves. Moreover, even though the responses to treatments showed clear family-level habitat preferences among linyphiids and lycosids, certain species within these families showed opposite responses. Thus, oversimplification of family-level habitat preferences should also be avoided.

What our results clearly demonstrate is an extremely slow recovery rate of the community after heavy grazing, which extends from the lichens even to invertebrates such as spiders. Both of our intermediate treatments with eight years of reindeer exclusion or eight years of grazing were strikingly similar to each other and to the long term grazed treatment. Thus we could not see signs of noticeable recovery from heavy grazing after eight years of protection from grazing, but on the other hand formerly ungrazed habitat was very similar to habitat that had been grazed for decades after eight years of grazing. The recovery of lichen pasture after heavy grazing is a long process (Olofsson, 2006; Hansen et al., 2007) and not only for slow-growing lichens, but also for other vegetation, the physical environment, and for faunistic components like the spiders studied here.

Supplemental Information

Supplemental Information 1 Complete list of specimens

Click here for additional data file.

We thank Tanja Kyykkä for her help in spider identification. Tapani Hopkins kindly checked the language.

Additional Information and Declarations

Competing Interests

Author Contributions

Field Study Permissions

Data Availability

The authors declare there are no competing interests.

Teemu Saikkonen performed the experiments, analyzed the data, prepared figures and/or tables, authored or reviewed drafts of the paper, approved the final draft.

Varpu Vahtera prepared figures and/or tables, authored or reviewed drafts of the paper, approved the final draft.

Seppo Koponen authored or reviewed drafts of the paper, approved the final draft.

Otso Suominen conceived and designed the experiments, performed the experiments, analyzed the data, prepared figures and/or tables, authored or reviewed drafts of the paper, approved the final draft.

The following information was supplied relating to field study approvals (i.e., approving body and any reference numbers):

The study sites are in the restricted border zone on Finnish side of the Finnish-Russian border. We had an official permit for the study from the Finnish Boarder Guard (Permit number 2717/2015).

The following information was supplied regarding data availability:

Raw data are available in the Supplemental File.

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
