# Peer review of "Effects of reindeer grazing and recovery after cessation of grazing on the ground-dwelling spider assemblage in Finnish Lapland"

_PeerJ, doi:10.7717/peerj.7330_

## Round 0.1 · original submission · Major Revisions

This manuscript is a valuable contribution to our understanding of reindeer grazing on northern ecosystems, since effects on invertebrates are large overlooked. However, both reviewers point out that the manuscript needs to be better organized and especially to clarify your hypothesis and to get a red thread between the hypotheses thorough out the manuscript to your conclusions. Reviewer 1 also has major critics on the statistical analysis and I think you need to address these issues.

Reviewer 1 ·

Basic reporting

Unfortunately the text is not well organized and there are problems with Figs 3 and 9. See my suggestions in General comments. The raw data are sorted by species and trap lines, some data based on individual pitfall traps are also needed.

Experimental design

The specific hypothesis is rather complicated and it needs some clarification. The response variables and sample sizes are not sufficiently well described. See my suggestions in General comments.

Validity of the findings

Parts of the statistical analysis is questionable and should be revised. See my suggestions in General comments.

Additional comments

I think this is an interesting study. Given the circumstances with fenced areas, it is possible to investigate the effects of reindeer grazing. This is a unique situation and I congratulate the authors to have taken the opportunity to explore these data. The authors are basing their conclusions on a large data set and they have accomplished data collection of a number of environmental variables that are pertinent to combine with the grazing effects. However, on the down-side of this study, there is a general weakness in the statistical analyses and the presentation of a massive number of results is making the text rather unstructured. I would also suggest that the authors clarify their "specific hypothesis" (Lines 88-93) as it is rather complicated.

My main criticism is focusing on the analysis of data.
- First, the authors are performing a very large number of tests on their data. A general rule of thumb is to include as much data as possible in each test in order to keep the number of tests at a minimum. There might be problems with the interpretation of p-values in the case of an excessive use of statistical tests. However, there are ways to reduce the number of tests. For instance, start by focusing on species and specimen numbers as response variables. It should be possible to perform ANCOVAs including both experimental treatment and some of the environmental variables. If this is not possible, the authors need to explain why. When the influence of the environment on spiders has been established, then continue with details in the ordination studies.
-Second, it could also be questioned why vegetation height is divided into several taxonomic components. These measurements (e.g. lingonberry, bilberry, etc) are analyzed separately for differences between experimental treatments in one-way ANOVAs. There are no attempts to link vegetation height, other than lichen, to the spiders. On line 246- it is claimed that a number of spider species are affected by "increasing lichen dry mass and height" (referring to Table 5): a large number of linear regressions on separate spider species and families are presented. Vegetation structure per se is since long known to affect spiders both at species and community level (see multiple studies over several decades, starting with e.g. Duffey 1962 Oikos 13; Colebourne 1974 J Anim Ecol 43; Hatley & McMahon 1980 Env Entomol 9, etc). Therefore, the focus on vegetation height of different plant taxa that is analyzed separately in the present manuscript can be questioned. I suggest that a general quantification of vegetation "height" and/or complexity could be useful to examine the effects of vegetation on the spider community.
- Third, the tests that are shown in Figure 9 and Table 4 are not based on independent data. The "total abundance" and "species richness" include the families being tested. From a statistical point of view this is not acceptable.
- Fourth, the exact response variables are not well described. For instance, it is unclear what "Species" and "Specimen" in Table 3 are referring to? What are the sample sizes? 180 pitfall traps? This means that the paragraph "Data analysis" (line 146) needs to be expanded to provide full details about analyses.

The general presentation of hypotheses, data and conclusions needs to be revised. I would suggest that the authors consider dropping some of the tests that are not essential to the "bigger picture" (e.g. "Evenness of Lycosidae and Linyphiidae spiders" (line 382), "Grazing effects on Thomisidae spiders" (line 395)). On line 434 the authors state: "Even though this study was not designed to reveal bioindicators..." – I suggest this part of the study to be omitted. Moreover, as the text is organized in the present version, it is very difficult to follow the line: hypotheses, methods, results, conclusions. Please, revise the manuscript accordingly.

Albeit the data can be regarded as the outcome of an "experiment", I think the authors should point out that the study is based on data from one site in one year. Addition of another site, or another year, would have made a great difference. Therefore, I suggest that the Conclusions (line 464 -) are revised with this reservation in mind.

Specific comments:
- Line 97: Nothing about the vegetation of the study area is presented under "Study area and methods". On line 467 ("Conclusions") it suddenly turns out that the study area is "lichen dominated dry pine forest". This information should be given in the beginning of the paper.
- Line 170: How was the conclusion about no "fence effect" reached?
- Fig 3: The labels on x-axis are not the same as in the figure legend.
- Fig 9: There are no labels on x- or y-axes.
- The data file is entirely sorted by species and trap lines. Data based on individual pitfall traps for species and specimen numbers should also be provided.

Reviewer 2 ·

Basic reporting

The article is written in good English. Literature references and background/context are insufficient for publication in their present form (see my general comments to the authors). Figures and tables are informative and well done.

Experimental design

Experimental design is appropriate, research question well defined and relevant, methods described in sufficient detail.

Validity of the findings

Findings are valid. Conclusions or the take-home message does not come out clearly.

Additional comments

Please see a separate attachment for detailed comments and suggestions for improvement.

Annotated reviews are not available for download in order to protect the identity of reviewers who chose to remain anonymous.

---

## Round 0.2 · Minor Revisions

The authors have done most of the job the revise this manuscript to a publishable standard, but to able to accept it they need to address the reviewers consider about:

1. The dependency between the spider families
2. Cite the important and basic research in the field

Furthermore, I believe the manuscript will be improved if authors added a figure showing the experimental design with traps and trap lines. Second, I think the vascular plant analysis would benefit of increase the resolution of this group into plant functional types, where more information for the reader will be available. Third, you also state that the main focus on this study is the Spiders (which I agree on), but the first section in the discussion give you a different impression. Please, rearrange the discussion so that your most important findings are discussed first. However, I believe all the revisions needed are of minor character. Finally, some specific suggestions:

L99-101: Is the disturbance really intermediate, is it just the time that past that are shorter?
L157-158: Please, rephrase this sentence. What do you mean with undergrowth and on the spot?
L160: Do you mean 5 cm in the soil?
L162: Which studies? Please give references.
L196-200: Could this be an effect of sampling, since you use a destructive sampling method?
L331: Spider instead of faunal?
L439-452: I believe this part is superfluous, and that you study isn’t design to address this. Please, delete.

Reviewer 1 ·

Basic reporting

no comment

Experimental design

no comment

Validity of the findings

no comment

Additional comments

I think the manuscript now is considerably improved.
But I still think the correlation tests between spider families (Table 4) are problematic because data are not independent. This could be solved quite easily by correlation test of e.g. Family 1 vs (All Families minus Fam 1), etc.

Reviewer 2 ·

Basic reporting

Review on manuscript “Effects of reindeer grazing and recovery after cessation of grazing on the ground-dwelling spider assemblage in Finnish Lapland” by Saikkonen et al.

In my opinion, authors did a fairly good job in revising this manuscript. Some issues remain, which I have listed below.

Line 36: Why call grazed area as a “degraded state” here? How does using this term reflect the main finding of the study (= no effect of grazing on spider abundances)? Provide a more thorough justification for using this term or revise it into something more neutral.

Line 47: “This is since…” does not fully reflect the reason why reindeer stay at the same areas throughout the year. The reindeer management system was originally created, because reindeer herds could no longer access the natural summer ranges.

Lines 93-96, and the rest of the paper. While I appreciate seeing some of these newer papers cited in this manuscript, these papers do not fit that well in the context of the paper. As the effects of grazing depend very heavily on the vegetation type as well as the grazing patterns, it would help concentrating on the previous work conducted in boreal systems with a lichen-rich vegetation, similar to the present study. Please, replace the sentence written in 93-96 with a sentence summarizing the effect of reindeer grazing on vegetation, nitrogen and carbon mineralization and soil carbon stocks in lichen-rich ecosystems. By this way, the reader may be able to link the present investigation on spider assemblages with the big picture.

Suggested reading:

Köster, E., K. Köster, M. Aurela, T. Laurila, F. Berninger, A. Lohila, and J. Pumpanen. 2013. Impact of reindeer herding on vegetation biomass and soil carbon content: a case study from Sodankylä, Finland. Boreal Environmental Research 18:35-42.

Köster, K., F. Berninger, E. Köster, and J. Pumpanen. 2015. Influences of reindeer grazing on above- and below-ground biomass and soil carbon dynamics. Arctic, Antarctic and Alpine Research 47:495-503.

Köster, K., E. Köster, L. Kulmala, F. Berninger, and J. Pumpanen. 2016. Are the climatic factors combined with reindeer grazing affecting the soil CO2 emissions in subarctic boreal pine forest? Catena 02838:dx.doi.org/10.1016/j.catena.2016.1006.1011.

Santalahti, M., Sund, H., Sietiö, O.-M., Köster, K., Berninger, F., Laurila, T., Pumpanen, J., Heinonsalo, J. 2018. Reindeer grazing alter soil fungal community structure and litter decomposition related enzyme activities in boreal coniferous forests in Finnish Lapland. Applied Soil Ecology 132:74-82.

Stark, S., Männistö, M.K., Smolander, A. (2010): Multiple effects of reindeer grazing on soil processes in nutrient-poor boreal forests. Soil Biology & Biochemistry 42: 2068-2077.

Susiluoto, S., T. Rasilo, J. Pumpanen, and F. Berninger. 2008. Effects of grazing on the vegetation structure and carbon dioxide exchange of a Fennoscandian fell ecosystem. Arctic, Antarctic and Alpine Research 40:422-431.

Also a good review on the effects on vegetation: Bernes C, Bråthen KA, Forbes BC, Hofgaard A, Moen J, Speed JD. 2015. What are the impacts of reindeer/caribou (Rangifer tarandus L.) on arctic and alpine vegetation? A systematic review protocol. Environmental Evidence 2:6.

Line 814. Having four different grazing treatments is a very novel and interesting feature of the present study and the link with the intermediate disturbance hypothesis puts the study into a very interesting context. Here is one more suggestions for a relevant paper considering the role of slow lichen growth on the recovery of lichen vegetation after disturbance:

Kumpula, J., A. Colpaert, and M. Nieminen. 2000. Condition, potential recovery rate, and productivity of lichen (Cladonia spp.) ranges in the Finnish reindeer management area. Arctic 53:152-160.

Experimental design

no comment

Validity of the findings

no comment

---

## Round 0.3 · accepted · Accept

The authors have sufficiently addressed the comments by the reviewers and myself.